# Multi-site Benchmarking of Deep Learning Models for Intraparenchymal Hemorrhage Segmentation on NCCT

**Kauê T N Duarte**[*1] ⓘD
**Abhijot S Sidhu**[1,2,3] ⓘD
**Murilo C Barros**[4] ⓘD
**Taha Aslan**[6] ⓘD
**Donghao Zhang**[5] ⓘD
**Jianhai Zhang**[1] ⓘD
**Devansh Bhatt**[1]
**Brij Karmur**[1] ⓘD
**Mohamed AlShamrani**[6] ⓘD
**Wu Qiu**[5] ⓘD
**Aravind Ganesh**[1] ⓘD
**Bijoy K Menon**[1] ⓘD

[1] *Cummings School of Medicine, University of Calgary, Calgary, AB, Canada.*

[2] *Graduate Program in Biomedical Engineering, University of Calgary, Calgary, Canada.*

[3] *Seaman Family MR Research Centre, Foothills Medical Centre, Calgary, Canada.*

[4] *School of Technology, University of Campinas, Limeira, Brazil.*

[5] *College of Life Science and Technology, Huazhong University of Science and Technology, Wuhan, China.*

[6] *Calgary Stroke Program, Department of Clinical Neurosciences, Foothills Medical Centre, University of Calgary, Calgary, Canada.*

**Editors:** Accepted for publication at MIDL 2026

## Abstract

Intraparenchymal hemorrhage (IPH) is a critical and often fatal subtype of hemorrhagic stroke, requiring rapid and accurate diagnosis on non-contrast computed tomography (NCCT) scans for effective treatment. While deep learning (DL) models, particularly convolutional neural networks (CNNs), offer potential for automating IPH segmentation, their real-world clinical utility is often limited by the lack of explicit data integration across diverse hospital sites with varying imaging protocols. This study conducted a multi-site benchmarking of five prominent CNN architectures: baseline U-Net, Attention U-Net, Feature Pyramid Network (FPN), Swin U-Net, and Trans U-Net, for IPH segmentation on a heterogeneous dataset from 17 clinical sites. Models were rigorously evaluated using F-measure (*a.k.a.*, Dice), Intersection over Union (IoU), and 95% Hausdorff Distance ($d_{H95}$). The advanced CNN variants (Attention U-Net, FPN, Trans U-Net) significantly outperformed the baseline U-Net in F-measure and IoU (*e.g.*, FPN F-measure: 0.868 vs. U-Net: 0.819, $p < 0.001$), with no significant difference among them. For boundary error, FPN reduced $d_{H95}$ compared to the baseline, whereas Trans U-Net showed improvement, though it was not significant. These models exhibited robust cross-site generalization across hemorrhage volumes, with minimal site-specific effects on performance. This study demonstrates that advanced

---

\* Corresponding author

CNN variants can be adopted for IPH segmentation to standardize and potentially accelerate IPH diagnosis.

**Keywords:** stroke, intraparenchymal hemorrhage, artificial intelligence, medicine, computed tomography.

## 1. Introduction

Stroke is a major cause of death and long-term disability globally. Each year, more than 12 million cases and over 7 million deaths are reported (Feigin et al., 2025). Among these cases, hemorrhagic stroke is one of the deadliest types, as it causes a rupture of cerebral blood vessels and subsequent intracranial bleeding. Although accounting for a smaller portion of the stroke cases, this type is associated with a high fatality rate.

Non-contrast computed tomography (NCCT) is a commonly used medical imaging modality for detecting stroke. It not only provides rapid, accessible information on intracranial hemorrhage (ICH) but also plays a critical role in emergency diagnosis and treatment planning. Fast detection of hemorrhage can positively salvage brain function and increase the patient's survival rate (Ahmed and Prakasam, 2025). This urgency, in a clinical setting, can affect the time to diagnosis, potentially leading to delays or oversights.

Among the subtypes of ICH, intraparenchymal hemorrhage (IPH) represents a critical and challenging pathology. IPH is characterized by bleeding in the brain tissue ($\sim$15% of total stroke cases), which leads to a high mortality rate (with 30-day mortality rates of 40-50%) among the ICH subtypes (Roy et al., 2015). This mortality rate is nearly double that of the fatality caused by ischemic stroke (Rothwell et al., 2004; Woo et al., 2022). A baseline hematoma volume is one of the strongest independent predictors of mortality, with patients with volumes $> 30mL$ experiencing a mortality rate $> 50\%$ (Abulhasan et al., 2023). This volume often drives therapeutic choices, such as selecting candidates for minimally invasive evacuation or deciding on surgery after follow-up imaging (Polster et al., 2021). Unlike other subtypes of ICH, like subdural or subarachnoid hemorrhage, where surgical evacuation is primarily anatomically guided, IPH management often relies heavily on precise volumetric quantification. However, accurately measuring IPH is challenging due to irregular lesion boundaries, variable texture patterns, and proximity to complex anatomical structures, all of which can affect measurements. These typically require specialized, robust analytical tools to segment and measure these regions in different sites.

Artificial Intelligence (AI) techniques have accelerated stroke diagnosis by automating manual tasks, such as detection and segmentation, while maintaining high accuracy rates. Among AI types, convolutional neural networks (CNNs) automatically extract information from images and are primarily used for classification tasks, such as ASPECTs scoring. For semantic segmentation, CNN-based U-Net variants and their numerous adaptations are often adopted, as they build an encoder-decoder architecture that not only extracts features but also reconstructs them in image space (Lin et al., 2025). These models have demonstrated high confidence in distinguishing pathological tissue from healthy tissue (Duarte et al., 2024). The advanced U-Net and its variants are continually improving, either by integrating attention mechanisms to delineate lesion boundaries more accurately or by using fractal pyramid networks to capture fine-grained and global contextual details.

One major challenge for clinical translation is domain shift across sites, which introduces variability across scanner vendors, acquisition protocols, and site practices. Variability in AI contexts can degrade model performance if not adequately tested in real-world clinical settings. The literature often refers to models trained on curated datasets, yet lacks systematic, comparative validation of these architectures for IPH segmentation across multiple clinical sites. Additionally, numerous studies have trained models on public data, addressing multiple hemorrhage types simultaneously, rather than optimizing for the complexities of IPH individually (Ahmed and Prakasam, 2025; Piao et al., 2023). The focus on architectural novelty can also overshadow deeper investigation of IPH's intrinsic features.

We propose a study focused on IPH segmentation across multiple sites. We benchmarked five CNN architectures (U-Net, Attention U-Net, FPN, Swin U-Net, and Trans U-Net) on a heterogeneous, multi-site NCCT dataset and report F-measure, IoU and $d_{H95}$. We evaluate generalizability by assessing statistical values using several metrics.

The remainder of this paper is organized as follows. Section 2 reviews related work relevant to the study. Section 3 details the materials, methods, and statistical definitions employed. Section 4 presents the results, and Section 5 provides an analysis of these findings. Finally, Section 6 outlines the conclusions and suggests directions for future research.

## 2. Related Work

The accurate and timely segmentation of intracranial hemorrhage on NCCT is essential for acute stroke management, impacting diagnostic speed, treatment planning, and, ultimately, patient outcomes (Ahmed and Prakasam, 2025). However, manual interpretation by radiologists can be time-consuming and is often subject to inter-observer variability (Inkeaw et al., 2022). Thus, deep learning models can solve this by automating segmentation, thereby reducing diagnostic delays and standardizing analysis (Piao et al., 2023).

In response to pressing clinical needs, researchers have concentrated on developing advanced segmentation architectures. Models such as the U-Net framework and its variants, U-Net++, Attention U-Net, and ResU-Net, have demonstrated strong performance on curated public benchmarks (Lin et al., 2025). In a systematic review, Zarei *et al.* (Zarei et al., 2024) corroborated that U-Net-based architectures are particularly powerful for IPH segmentation on NCCT. More recently, transformer-based models and hybrid architectures, such as TransHarDNet, have been explored to more effectively capture long-range dependencies (Piao et al., 2023). Aside from segmentation tasks, studies such as Gong *et al.* (Gong et al., 2023) proposed unified frameworks for simultaneous ICH volume quantification (regression) and patient prognosis, using a feature extractor based on 3D ResNet and adopting Grad-CAM for enhanced interpretability. These models consistently achieve high Dice Similarity Coefficients, sometimes reaching 0.85 on their respective test sets (Ahmed and Prakasam, 2025; Lin et al., 2025), highlighting the considerable potential of deep learning for this task. For multiple downstream tasks, Zhang *et al.*(Zhang et al., 2025) proposed a multi-task study with the focus of understanding the use of DL for several hemorrhage applications. However, when a large amount of data is available, authors such as Gerbasil *et al.* (Gerbasi et al., 2025) decide to integrate foundation models. They introduced a semi-automated pipeline for IPH segmentation, combining a fine-tuned YOLOv8-S model for slice-specific lesion detection with a prompted Medical Segment Anything Model.

Still focused on multiple downstream tasks, Kaur *et al.*(Kaur and Singh, 2024) introduced models to detect, segment, and classify multiple ICH subtypes, including IPH, using a customized CNN for segmentation and a Hybrid YOLO-HD model for classification. When targeting the IPH specifically, Juan *et al.* (Juan et al., 2026) developed a multitask pipeline for classification, detection, and weakly supervised 3D segmentation. Their framework integrates parallel tasks for ICH classification using SE-ResNeXt-50, perihematomal edema detection with YOLOv8 and RT-DETR, and a novel 3D PHE-pretrained nnU-Net for IPH segmentation using pseudo-labels derived from edema masks.

However, this predictive performance is often obtained and validated using homogeneous or publicly available datasets collected with standardized imaging protocols (Roy et al., 2015). When implemented in real-world clinical settings, these models can potentially encounter a notable domain shift, resulting in lower performance (Inkeaw et al., 2022). This shift is driven by variations in scanner vendors, acquisition parameters (e.g., tube voltage and slice thickness), and reconstruction kernels across hospitals. Although some studies have begun to address this issue through approaches such as multi-window input optimization (Inkeaw et al., 2022), a gap remains in validating segmentation models on large, heterogeneous, multi-site datasets. The comparative analysis of how CNN variants can generalize across multiple clinical sites remains underexplored, despite its vital importance for real-world applications.

This validation paper studies deep learning models for IPH segmentation using a multi-site dataset characterized by significant protocol heterogeneity. The model was designed explicitly for segmenting parenchymal hemorrhage. Unlike existing studies, our work uniquely quantifies the performance and generalization of this specialized model across a diverse, multi-hospital private dataset, rather than simply developing a new architecture based on public data or addressing a wide array of hemorrhage types. This approach provides a crucial real-world evaluation of the challenges of AI deployment in stroke care, particularly highlighting how different architectural strategies maintain performance across varied clinical settings.

## 3. Materials and Methods

### 3.1. Dataset and Participant Demographics

The study utilized a multi-site dataset with IPH segmentation. In total, $N = 239$ subjects were included from 17 clinical sites across Canada (labelled A-Q in accordance with our ethics board) participating in the ACT Trial imaging collection (Menon et al., 2022). All imaging data included manual ground-truth annotations for intraparenchymal hemorrhage (IPH), along with descriptive information such as age, sex, and other factors. Table 1 summarizes the demographic and clinical characteristics of the study population.

**ACT Trial Imaging Dataset**. We used CT imaging data and corresponding hemorrhage segmentation masks provided by the ACT Trial investigators (Menon et al., 2022). Site identifiers were anonymized in accordance with ethical and data-sharing agreements. Hemorrhagic stroke diagnoses were confirmed by board-certified neurologists using standardized clinical criteria. Ground-truth segmentation masks were generated using a semi-automated workflow that combines algorithmic lesion proposals with expert manual correction and review.

Table 1: Demographic and clinical characteristics of the study population across the 17 anonymized clinical sites (A-Q). Data are presented as Mean ± Standard Deviation for Age (years) and IPH Volume (cm³). Sex is reported as the count of male patients with the corresponding percentage in parentheses. The sample size (N) for each site is also provided.

|  | **A** | **B** | **C** | **D** | **E** |
|---|---|---|---|---|---|
| Age | 71.37 ± 15.00 | 59.00 ± 18.38 | 72.97 ± 13.74 | 78.50 ± 12.07 | 78.78 ± 12.62 |
| Sex | 30 (55.6%) | 2 (100.0%) | 21 (52.5%) | 4 (50.0%) | 5 (55.6%) |
| Vol | 12.69 ± 22.81 | 24.54 ± 32.42 | 25.29 ± 50.69 | 19.18 ± 28.48 | 3.32 ± 4.37 |
| N | 54 | 2 | 40 | 8 | 9 |

|  | **F** | **G** | **H** | **I** | **J** |
|---|---|---|---|---|---|
| Age | 70.57 ± 12.14 | 75.75 ± 11.57 | 74.17 ± 9.39 | 72.60 ± 19.50 | 72.43 ± 12.99 |
| Sex | 3 (42.9%) | 3 (37.5%) | 2 (33.3%) | 2 (40.0%) | 18 (64.3%) |
| Vol | 3.54 ± 3.09 | 17.05 ± 36.22 | 10.17 ± 14.15 | 7.66 ± 12.61 | 7.50 ± 17.22 |
| N | 7 | 8 | 6 | 5 | 28 |

|  | **K** | **L** | **M** | **N** | **O** |
|---|---|---|---|---|---|
| Age | 76.64 ± 12.18 | 80.89 ± 11.89 | 84.50 ± 6.81 | 73.58 ± 11.68 | 77.33 ± 7.51 |
| Sex | 2 (18.2%) | 7 (77.8%) | 8 (50.0%) | 12 (46.2%) | 2 (66.7%) |
| Vol | 4.87 ± 5.42 | 34.84 ± 40.85 | 18.71 ± 32.98 | 12.25 ± 28.40 | 6.88 ± 8.50 |
| N | 11 | 9 | 16 | 26 | 3 |

|  | **P** | **Q** | **Total** |
|---|---|---|---|
| Age | 75.20 ± 15.32 | 87.50 ± 2.12 | 74.40 ± 13.29 |
| Sex | 3 (60.0%) | 2 (100.0%) | 126 (52.71%) |
| Vol | 0.85 ± 1.42 | 0.44 ± 0.56 | 14.27 ± 30.36 |
| N | 5 | 2 | 239 |

### 3.2. Data Preparation

To improve the quality of the NCCT scans, we applied skull stripping using SynthStrip, adjusted to CT (Hoopes et al., 2022). All 3D volumes were standardized to dimensions that are multiples of 64 through zero-padding. Each volume was split into $64 \times 64 \times 64$ patches to facilitate memory-efficient processing. To address class imbalance between hemorrhagic and non-hemorrhagic voxels, we employed a selective patching strategy that retained patches containing at least one hemorrhagic lesion voxel during training, validation, and testing. Intensity normalization was carried out in two steps: (1) we cropped the intensity from -30 to 100 Hounsfield units (HU); (2) we performed min-max normalization, mapping the image intensities to the range $[0.0, 1.0]$.

### 3.3. Deep Learning Architectures

Our benchmarking study focuses on the U-Net architectures. We designed using only U-Net architectures for two main purposes: (1) to validate and compare commonly deployed architectures in medical image segmentation, and (2) to draw broader conclusions about how specific architecture-controlled enhancements impact performance on an IPH segmentation. We utilized attention mechanisms, multi-scale feature fusion, and transformer-based context modelling, compared to the baseline U-Net. U-Nets use an encoder-decoder architecture with skip connections, enabling efficient feature localization. We intentionally excluded large foundation models (*e.g.*, prompt-driven SAT) and end-to-end self-supervised pretraining to keep the comparison controlled and reproducible, since pretraining introduces orthogonal variables (pretraining data/objective, prompt/adapter design) and typically requires substantially larger unlabeled data to compute.

   We implemented and compared five state-of-the-art CNN variants for IPH segmentation:

1. *Baseline U-Net*: We implemented the original U-Net architecture (Ronneberger et al., 2015) as our baseline model. This encoder-decoder network with skip connections provides a robust foundation for medical image segmentation.

2. *Attention U-Net*: This architecture enhances the traditional U-Net by incorporating attention gates in the skip connections (Oktay et al., 2018). The attention mechanisms selectively emphasize relevant spatial features while suppressing irrelevant regions, particularly beneficial for detecting small hemorrhagic lesions and precise boundary delineation.

3. *Feature Pyramid Network (FPN)*: The FPN architecture (Lin et al., 2017) builds a multi-scale feature pyramid through top-down pathways and lateral connections. This design enables effective feature extraction across multiple scales, which is advantageous for detecting hemorrhagic lesions of varying sizes and shapes.

4. *Trans U-Net*: This architecture leverages a hybrid of a CNN+Transformer design (Chen et al., 2021), combining U-Net local feature extraction and the Vision Transformer (ViT). The integration of ViT and U-Net yields improved global context for IPH masks and is believed to enhance boundary delineation.

5. *Swin U-Net*: This architecture (Zhang et al., 2023) introduces a hierarchical Swin Transformer as the encoder to capture long-range dependencies across image patches. A symmetric Swin Transformer-based decoder, connected via skip connections, then upsamples the features to generate segmentation maps.

All architectures utilized a VGG16 backbone (Simonyan and Zisserman, 2014) for feature extraction in the encoder pathway, consistent with previous work demonstrating its effectiveness for medical image segmentation tasks (Duarte et al., 2024, 2022).

### 3.4. Model Training and Evaluation

Model training was conducted for a maximum of 300 epochs with an initial learning rate of $5 \times 10^{-4}$. We employed the Adam optimizer and reduced the learning rate on plateau using Keras' `ReduceLROnPlateau`. Specifically, we monitored validation IoU and used the following settings: `factor=0.5`, `patience=8`, `mode='max'`, `min_lr=`$1 \times 10^{-7}$, `cooldown=0`, and `verbose=1`. Thus, when the monitored validation IoU did not improve for 8 consecutive epochs, the learning rate was multiplied by 0.5, down to a minimum of $1 \times 10^{-7}$. The model with the highest validation IoU was saved as the final model. For each architecture, we trained separate 2D models using axial (2DAxi), coronal (2DCor), and sagittal (2DSag) projections, and obtained final predictions by averaging across these projections (2.5D model).

**Loss Function**. To address the class imbalance between *True* and *False* elements in the IPH masks, we used a composite loss function combining Dice Loss (*DL*, eq. 1) and Binary Focal Loss (*FL*, eq. 2).

$$DL = \frac{(1 + \beta^2) \cdot TP}{(1 + \beta^2) \cdot TP + \beta^2 \cdot FN + FP} \tag{1}$$

where $\beta$ corresponds to a balance coefficient, *TP*, *FP*, and *FN* represent the true positive, false positive, and false negative voxels, respectively. For all reported experiments, we set $\beta = 1.0$ (equal weighting of precision and recall).

$$FL = -GT\alpha(1 - PT)^\gamma \log(PT) \quad -(1 - GT)\alpha PT^\gamma \log(1 - PT) \tag{2}$$

where *GT* is the ground-truth values, *PT* represents the predicted truth, $\alpha = 0.25$ and $\gamma = 2.0$ are values that were fine-tuned through a calibration process.

**Performance Metrics**. The class imbalance between IPH and non-IPH voxels rendered accuracy an unsuitable performance metric, as the large number of true negatives (*TN*) would disproportionately influence the results. To better evaluate model performance, we utilized the *F*-measure, intersection-over-union (IoU), and Hausdorff distance. The *F*-measure, *a.k.a.* dice coefficient for binary segmentation, is a commonly adopted metric in image segmentation:

$$F = 2 \times \frac{P \times R}{P + R} \tag{3}$$

which represents the harmonic mean of precision (*P*) and recall (*R*):

$$P = \frac{TP}{TP + FP} \ \text{ and } \ R = \frac{TP}{TP + FN}.$$

IoU quantifies the overlap between predictions and ground truth:

$$\text{IoU} = \frac{TP}{FP + TP + FN}. \tag{4}$$

Throughout training, the model achieving the highest IoU value was retained as optimal.

The Hausdorff distance measures the separation between the predicted and ground-truth IPH boundaries. For two point sets $A$ and $B$ representing these boundaries, the $d_{H95}$ is defined as:

$$d_H(x,y) \ = \ \max\{d_{AB}, d_{BA}\} \ = \ \max\left\{ \max_{a \in A}\left\{ \min_{b \in B}\{d(a,b)\}\right\}, \max_{b \in B}\left\{ \min_{a \in A}\{d(a,b)\}\right\}\right\} \tag{5}$$

where $a$ and $b$ represent elements of sets $A$ and $B$, respectively, and $d(a,b)$ is the Euclidean distance between them. We used the $95^{th}$ percentile of the Hausdorff distance distribution ($d_{H95}$) to assess performance. Superior performance is indicated by higher $F$-measure and IoU values, and a lower $d_{H95}$ value.

**Implementation Details**. We ran experiments on a four-node cluster ($8\times$ Tesla V100 16GB GPUs; 754 GB total system RAM. To enable a fair comparison with other U-Net variants, each brain projection was trained independently in parallel, substantially reducing the total training time. Model development was carried out using Python 3.6 in Jupyter Notebook, and the resulting code was subsequently converted to Python scripts to enable cluster execution. The full source code and Keras-based implementations are publicly available [1].

### 3.5. Statistical Analysis

Five-fold cross-validation was used to evaluate all five U-Net models, and results are reported as the mean $\pm$ standard deviation. Throughout the analysis, appropriate tests were performed to assess the validity of the model assumptions, including tests of residual normality. Significance level was set to $\alpha = 0.05$ for all statistical tests. The R statistical package (https://www.r-project.org/) was used.

To evaluate the effect of segmentation architecture on performance metrics PM(F-measure, IoU, and $d_{H95}$), we employed a two-stage linear modelling framework designed to account for substantial imbalance in sample sizes across acquisition sites. All analyses were performed separately for each segmentation architecture. First, to assess performance as a function of stroke severity in sites with limited sample sizes, we fitted a linear regression model aggregating sites with fewer than 15 cases. In this model, performance metrics were modelled as a function of ground-truth intraparenchymal hemorrhage volume ($IPH_{vol}$), which served as a surrogate measure of lesion severity. Site was not included as a covariate in this model due to insufficient within-site variability and the instability of site-specific estimates in small-sample settings. The model structure was:

---

1. https://github.com/KaueTND/ip-hemorrhagic-stroke-segmentation

$$PM \sim IPH_{vol} \tag{6}$$

Second, to evaluate algorithm performance and generalizability across acquisition sites with adequate sample sizes, a separate linear regression model was fitted, including only sites with more than 15 cases. In this model, site was included as a categorical fixed effect to account for acquisition-related variability, alongside hemorrhage volume and demographic covariates:

$$PM \sim IPH_{vol} + Age + Sex + Sitename \tag{7}$$

This model was used to assess whether segmentation performance varied systematically across sites under conditions in which site-specific effects could be reliably estimated.

To test for overall differences in mean performance across segmentation architectures (U-Net, Attention U-Net, FPN, Swin U-Net, and Trans U-Net), a one-way analysis of variance (ANOVA) was performed for each performance metric. When the omnibus ANOVA indicated a significant effect of segmentation architecture, post-hoc pairwise comparisons were conducted using t-tests with Bonferroni correction to control for multiple comparisons.

### 3.6. Clinically-specific Analysis

Performance analyses were stratified by hemorrhage volume and anatomical adjacency (sulci, ventricles, and others) to ensure model comparisons reflect clinical and methodological variability. IPH volume was categorized as small (<5 mL), moderate (5–30 mL), or large (>30 mL) because lesion size can potentially affect voxel-wise learning and error interpretation. In small IPH volumes, a single voxel misclassification can lead to a large relative volumetric error and significantly affect clinical severity estimates. For large IPH volumes, the same error can have less effect on volume estimates but may still influence boundary precision. We reported F-Measure and IoU for volumetric overlap, and $d_{H95}$ for boundary delineation.

### 4. Results

### 4.1. Segmentation Performance Across Architectures

Segmentation performance was evaluated using F-measure, Intersection-over-Union (IoU), and the 95th percentile Hausdorff distance ($d_{H95}$). Analyses were conducted separately for sites with limited sample sizes and sites with adequate sample sizes to account for substantial imbalance in site-level representation.

**Severity-dependent performance in small-sample sites.** For sites with fewer than 15 cases, performance was evaluated using linear regression models relating each performance metric to ground-truth intraparenchymal hemorrhage volume (*IPH_vol*), which served as a surrogate measure of lesion severity. Demographic variables and site were not included in these models due to insufficient within-site variability and the instability of site-specific estimates in small-sample settings.

Across all segmentation architectures, increasing *IPH_vol* was associated with improved overlap-based performance. Specifically, significant positive associations between *IPH_vol*

and F-measure were observed for all models (U-Net: $\beta = 1.98 \times 10^{-6}$, $p < 0.001$; Attention U-Net: $\beta = 1.37 \times 10^{-6}$, $p < 0.001$; FPN: $\beta = 1.42 \times 10^{-6}$, $p = 0.001$; Trans U-Net: $\beta = 1.41 \times 10^{-6}$, $p = 0.001$; Swin U-Net: $\beta = 2.30 \times 10^{-6}$, $p = 0.009$). Similar severity-dependent improvements in IoU were observed across all architectures (all $p < 0.01$).

For boundary accuracy, no significant associations between *IPH_vol* and $d_{H95}$ were observed in the small-site cohort across architectures (all $p > 0.16$), reflecting the limited sensitivity of boundary-based metrics in sparse, heterogeneous samples.

**Generalizability across large-sample sites.** For sites with at least 15 cases, segmentation performance was assessed using linear regression models that included *IPH_vol*, age, sex, and site as fixed effects. In these models, *IPH_vol* remained a significant predictor of performance across all architectures and metrics. Larger hemorrhage volumes were consistently associated with higher F-measure and IoU values (all $p < 10^{-4}$) and lower $d_{H95}$ values (all $p < 0.05$), indicating improved overlap and boundary accuracy for larger lesions.

No significant effects of age, sex, or acquisition site were observed for any performance metric after accounting for multiple comparisons (all $p > 0.05$). Although isolated site-specific coefficients reached nominal significance in uncorrected analyses, these effects did not survive correction for multiple testing and were not consistent across segmentation architectures or metrics. Collectively, these findings indicate that segmentation performance was not systematically influenced by demographic factors or acquisition site, supporting the robustness and generalizability of all models across heterogeneous clinical imaging conditions.

**Comparison of segmentation architectures.** To assess overall differences in mean performance across segmentation architectures, one-way analyses of variance (ANOVA) were conducted on the full dataset for each performance metric. Significant main effects of architecture were observed for all metrics, including F-measure ($F(4, 1190) = 7.17$, $p = 1.06 \times 10^{-5}$), IoU ($F(4, 1190) = 7.84$, $p = 3.1 \times 10^{-6}$), and $d_{H95}$ ($F(4, 1189) = 6.17$, $p = 6.51 \times 10^{-5}$).

Post-hoc pairwise comparisons with Bonferroni correction revealed that Attention U-Net, FPN, and Trans U-Net significantly outperformed the baseline U-Net for both F-measure and IoU (all adjusted $p < 0.01$). Swin U-Net demonstrated significantly lower overlap performance compared to the other architectures (all adjusted $p < 0.01$). No significant differences were observed among the three advanced convolutional architectures (Attention U-Net, FPN, and Trans U-Net) after correction for multiple comparisons.

For $d_{H95}$, FPN achieved significantly lower boundary error relative to the baseline U-Net ($p = 0.0287$), whereas no significant differences were observed between U-Net and Attention U-Net ($p = 0.118$) or Trans U-Net ($p = 1.000$). In contrast, Swin U-Net exhibited significantly higher boundary error compared to Attention U-Net ($p = 0.0017$) and FPN ($p = 0.00024$), while differences relative to Trans U-Net ($p = 0.228$) and U-Net ($p = 1.000$) were not significant. Mean performance values for each metric and architecture are reported in Tables 2, 3, and 4.

**Clinically relevant stratification.** Figure 3 illustrates F-measure performance stratified by acquisition site and segmentation architecture. In addition, Figure 4 evaluates segmentation performance across clinically relevant stratification, including anatomical hemorrhage

location (e.g., near sulci, ventricles, or other regions) and hemorrhage volume categories (<5 mL, 5–30 mL, and >30 mL), further demonstrating consistent performance trends across lesion characteristics.

## 5. Discussion

In this study, we compared and evaluated different CNN variants for IPH segmentation, with a particular focus on generalizability across multi-site clinical data. Performance was assessed using three commonly reported metrics in the literature (F-measure, IoU, and $d_{H95}$). Using a two-stage linear modelling framework followed by ANOVA-based architecture comparisons, we found that Attention U-Net, FPN, and Trans U-Net significantly outperformed the baseline U-Net in terms of volumetric overlap, as measured by F-measure and IoU (ANOVA: F-measure $F(4, 1190) = 7.17$, $p = 1.06 \times 10^{-5}$; IoU $F(4, 1190) = 7.84$, $p = 3.1 \times 10^{-6}$). In contrast, Swin U-Net demonstrated significantly lower overlap performance relative to the other architectures (Bonferroni-corrected $p < 0.01$ for all pairwise comparisons with Attention U-Net, FPN, and Trans U-Net).

For boundary detection, architectural differences were more selective. FPN achieved a statistically significant reduction in $d_{H95}$ relative to the baseline U-Net (Bonferroni-corrected $p = 0.0287$), whereas Attention U-Net ($p = 0.118$) and Trans U-Net ($p = 1.000$) did not show significant improvements. Swin U-Net exhibited significantly higher boundary error compared to Attention U-Net ($p = 0.0017$) and FPN ($p < 0.001$), while differences relative to Trans U-Net ($p = 0.228$) and U-Net ($p = 1.000$) were not statistically significant. These patterns are summarized in Tables 2–4. The 2DCor orientation achieved higher F-measure and IoU scores, whereas the 2.5D approach yielded lower $d_{H95}$ values, indicating improved boundary alignment with ground truth.

Across sites with adequate sample sizes, the advanced convolutional architectures (Attention U-Net, FPN, and Trans U-Net) demonstrated comparable performance in F-measure and IoU, with no statistically significant differences observed after correction for multiple comparisons (all Bonferroni-corrected $p \geq 0.684$). Swin U-Net, however, consistently underperformed these models across both overlap-based metrics (all Bonferroni-corrected, $p < 0.01$). These findings indicate that while multiple architectures achieve robust volumet-

Table 2: F-Measure for IPH Segmentation. Performance is compared across four CNN variants (Attention U-Net, baseline U-Net, FPN, Swin U-Net, and Trans U-Net) on axial (2DAxi), coronal (2DCor), and sagittal (2DSag) projections, and their ensemble (2.5D). Results are reported as Mean ± Standard Deviation. The best-performing model for each orientation is highlighted in **bold**.

| Style Orientation | Attention U-Net | FPN | Swin U-Net | Trans U-Net | U-Net |
|---|---|---|---|---|---|
| 2DAxi | $0.609 \pm 0.244$ | $0.635 \pm 0.225$ | $\mathbf{0.649 \pm 0.170}$ | $0.629 \pm 0.228$ | $0.617 \pm 0.221$ |
| 2DCor | $0.865 \pm 0.129$ | $\mathbf{0.868 \pm 0.127}$ | $0.845 \pm 0.135$ | $0.863 \pm 0.132$ | $0.819 \pm 0.153$ |
| 2DSag | $0.849 \pm 0.155$ | $\mathbf{0.854 \pm 0.140}$ | $0.822 \pm 0.150$ | $0.847 \pm 0.145$ | $0.811 \pm 0.164$ |
| 2.5D | $0.851 \pm 0.165$ | $\mathbf{0.862 \pm 0.146}$ | $0.842 \pm 0.139$ | $0.855 \pm 0.151$ | $0.823 \pm 0.164$ |

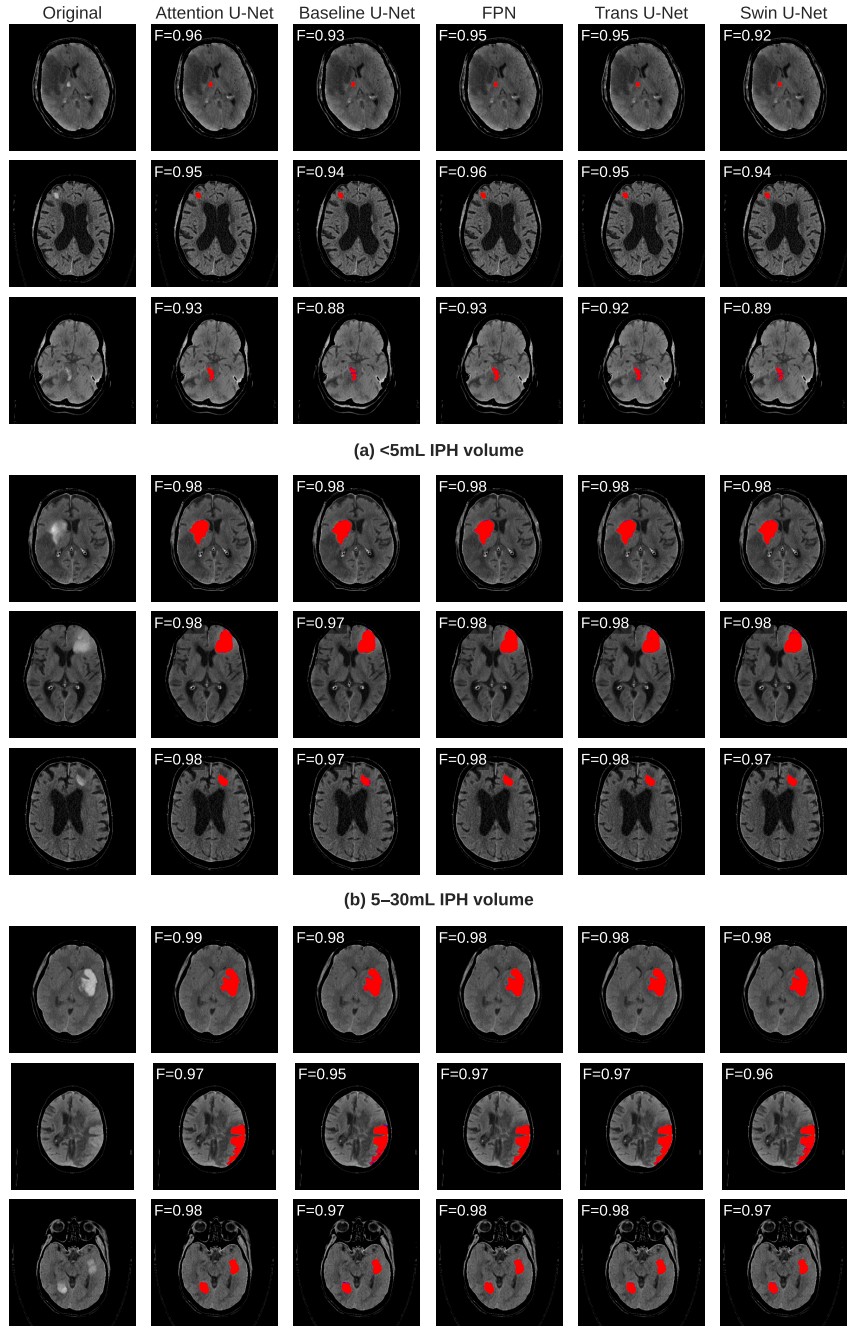

Figure 1: Qualitative comparison of IPH segmentation results across different CNN variants. Three IPH volume categories are shown: (a) <5 mL IPH volume, (b) 5–30 mL IPH volume, and (c) >30 mL IPH volume. Within each volume category, the first row corresponds to IPH located near the ventricles, the second row to IPH near the sulci, and the third row to IPH in other anatomical regions. F-Measure value is reported in the top left corner. Red, blue, and green colors corresponds to TP,FP,FN, respectively. Masks are overlaid in the original volume (no skull-stripped) for better visualization. 12

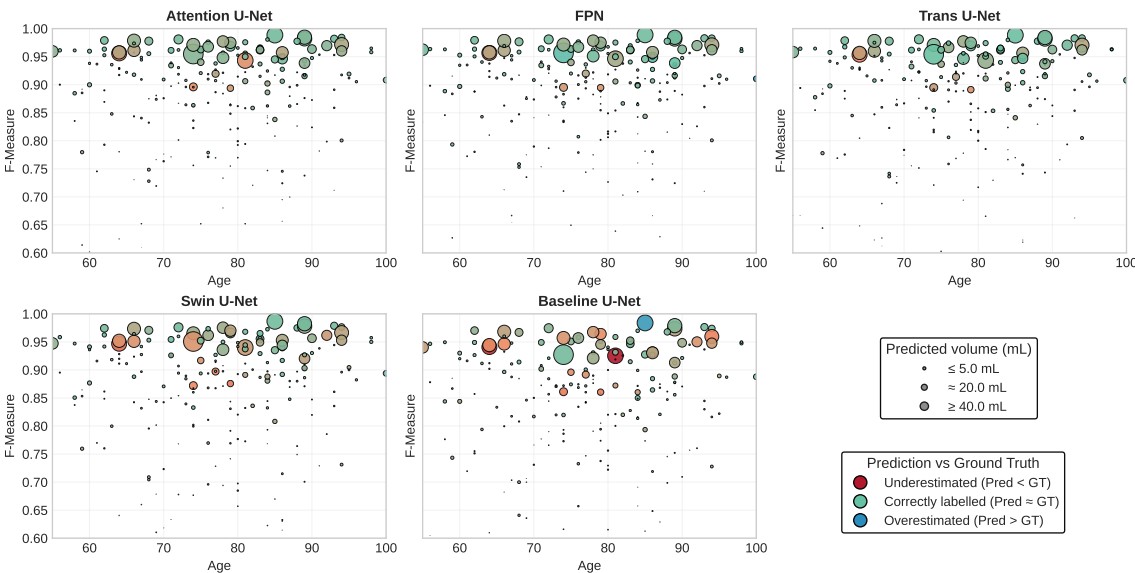

Figure 2: Scatter plot of subject age (x-axis) and F-Measure (y-axis) for IPH segmentation. Each point represents an individual subject, with marker size proportional to the IPH volume ($mL$) computed from the model's predicted segmentation mask. Colours go from red (underestimated according to ground-truth) → green (correctly labelled) → blue (overestimation compared to ground-truth).

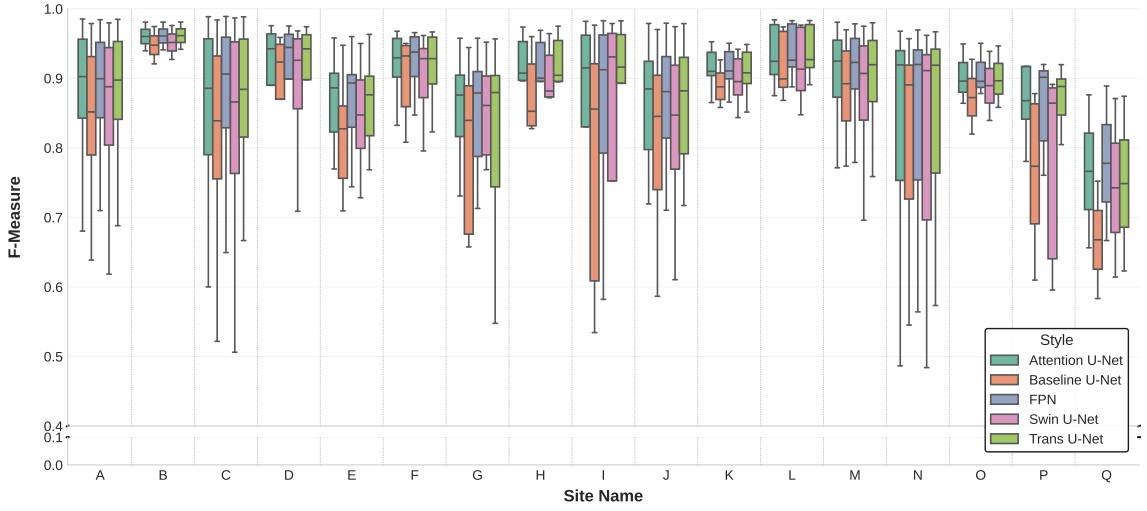

Figure 3: Boxplot comparison of F-Measure scores for IPH segmentation across the 17 clinical sites (A-Q), stratified by CNN variant.

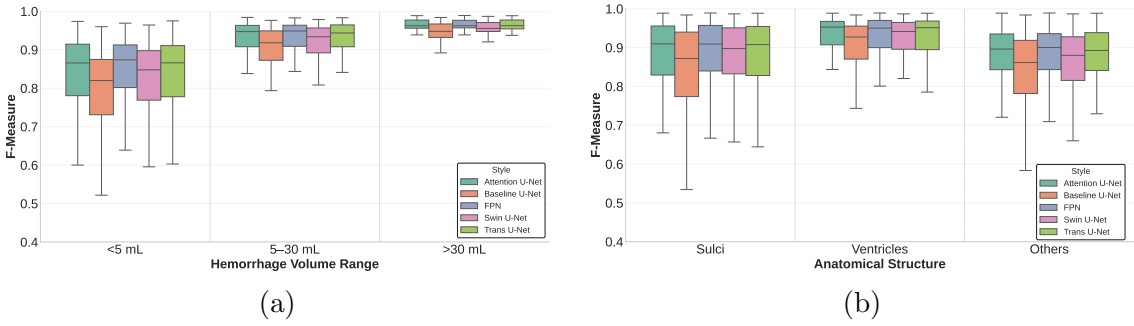

Figure 4: Boxplot comparison of F-Measure scores for IPH segmentation across CNN variants. (a) Performance stratified by hemorrhage volume ranges: small (<5 mL), medium (5–30 mL), and large (>30 mL). (b) Performance stratified by IPH anatomical location, grouped as lesions adjacent to sulci, ventricles, or other brain regions. Each box represents the distribution of subject-level F-Measure values for a given CNN variant within each subgroup.

ric overlap across heterogeneous acquisition conditions, transformer-based modelling alone does not guarantee improved performance for IPH segmentation in this setting.

In the large-site cohort, no statistically significant effects of age, sex, or acquisition site were observed for any performance metric after correction for multiple comparisons (all corrected $p > 0.05$). Although isolated site-level coefficients reached nominal significance in uncorrected analyses, these effects were inconsistent across architectures and metrics and did not survive correction. This absence of systematic demographic or site effects supports the robustness and generalizability of the evaluated models across heterogeneous clinical imaging conditions.

Volume- and anatomy-stratified analyses (Figure 4) further contextualized these findings. Small hemorrhages (<5 mL) posed the greatest challenge for both overlap-based and boundary-based metrics, whereas moderate (5–30 mL) and large (>30 mL) hemorrhages exhibited reduced variability and improved performance. These patterns align with the observed positive associations between IPH volume and F-measure/IoU and the neg-

Table 3: IoU for IPH Segmentation. Performance is compared across four CNN variants (Attention U-Net, baseline U-Net, FPN, Swin U-Net, and Trans U-Net) on axial (2DAxi), coronal (2DCor), and sagittal (2DSag) projections, and their ensemble (2.5D). Results are reported as Mean ± Standard Deviation. The best-performing model for each orientation is highlighted in **bold**.

| Style | Attention U-Net | FPN | Swin U-Net | Trans U-Net | U-Net |
|---|---|---|---|---|---|
| Orientation | | | | | |
| 2DAxi | $0.479 \pm 0.236$ | $0.502 \pm 0.225$ | $\mathbf{0.504 \pm 0.189}$ | $0.496 \pm 0.227$ | $0.481 \pm 0.221$ |
| 2DCor | $0.780 \pm 0.164$ | $\mathbf{0.785 \pm 0.161}$ | $0.751 \pm 0.176$ | $0.777 \pm 0.167$ | $0.718 \pm 0.187$ |
| 2DSag | $0.762 \pm 0.182$ | $\mathbf{0.767 \pm 0.173}$ | $0.721 \pm 0.189$ | $0.757 \pm 0.179$ | $0.709 \pm 0.194$ |
| 2.5D | $0.767 \pm 0.192$ | $\mathbf{0.779 \pm 0.176}$ | $0.749 \pm 0.177$ | $0.770 \pm 0.182$ | $0.726 \pm 0.194$ |

ative trends observed for $d_{H95}$. Importantly, the boundary-precision advantage of FPN persisted across volume and anatomical regions, whereas Swin U-Net consistently demonstrated higher boundary error across these clinically relevant subgroups. Segmentation performance improved with increasing hemorrhage volume across all models. Ground-truth IPH volume showed significant positive associations with F-measure and IoU across architectures (all $p < 0.001$), reflecting improved overlap for larger lesions. Associations between IPH volume and $d_{H95}$ were weaker and model-dependent; while negative trends were observed, these did not consistently survive correction for multiple comparisons. Collectively, these results indicate that all models perform better on larger, more confluent hemorrhages, which are inherently easier to segment.

The examination of the volume- (Figure 4.a) and anatomic-stratified (Figure 4.b) results aligned with our statistical findings. Smaller IPH volumes are more challenging for voxel-wise overlap (*e.g.*, F-Measure, and IoU) and boundary metrics ($d_{H95}$). This explains the larger variability in our metric evaluation. Moderate to large IPH volumes tend to have greater overlap, which helps smooth potential boundary errors. There is a positive association between IPH volume and F-Measure/IoU and a negative association with $d_{H95}$. While examining the anatomical context, we noticed that IPH volumes adjacent to cortical sulci pose additional segmentation challenges due to potential partial-volume effects, whereas hemorrhages near the ventricles tend to exhibit more consistent contrast, resulting in stable performance across models.

Our multi-site validation addresses a critical gap in previous segmentation studies, which were often limited to single-institution or publicly available datasets (Inkeaw et al., 2022). The consistent performance across sites suggests that the advanced architectures learn feature representations that are robust to site-specific variations in imaging protocols, making them suitable for broader clinical deployment.

When contextualized within the broader literature, our multi-site results demonstrate competitive, if not superior, performance and generalizability. While (Inkeaw et al., 2022) reported a median Dice coefficient of 0.37 for IPH segmentation (in multi ICH segmentation) and (Lin et al., 2025) achieved Dice scores around 0.91 for cerebral contusion segmentation, our FPN model achieved an F-measure of 0.868 (in Dice metric) while demonstrating robust multi-site performance. The CNN variant efficiencies of models like FPN, Attention U-Net, and Trans U-Net suggest a promising path toward developing solutions that are both highly

Table 4: $d_{H95}$ for IPH Segmentation. Performance is compared across four CNN variants (Attention U-Net, baseline U-Net, FPN, Swin U-Net, and Trans U-Net), evaluated on axial (2DAxi), coronal (2DCor), sagittal (2DSag) projections, and their ensemble (2.5D). Results are reported as Mean ± Standard Deviation. Values are in $mm$. The best-performing model for each orientation is highlighted in **bold**.

| Style | Attention U-Net | FPN | Swin U-Net | Trans U-Net | U-Net |
|---|---|---|---|---|---|
| Orientation | | | | | |
| 2DAxi | **10.066 ± 9.643** | 11.624 ± 8.507 | 11.907 ± 7.785 | 10.371 ± 8.964 | 11.465 ± 9.336 |
| 2DCor | 3.376 ± 9.066 | **2.916 ± 7.464** | 6.265 ± 10.987 | 4.855 ± 12.558 | 5.862 ± 11.051 |
| 2DSag | 4.578 ± 11.038 | **4.203 ± 10.113** | 9.488 ± 15.439 | 5.594 ± 10.934 | 6.871 ± 10.526 |
| 2.5D | 1.600 ± 4.974 | **1.574 ± 4.550** | 2.676 ± 8.557 | 1.615 ± 4.837 | 2.542 ± 6.356 |

accurate and computationally feasible for real-time use in emergency settings across multiple healthcare institutions (Piao et al., 2023).

In clinical practice, segmentation accuracy is more than just a technical measure; it directly affects important decisions. Errors in IPH segmentation can impact how we estimate hematoma volume, assess midline shift, measure edema, and track changes over time. These steps are crucial for predicting outcomes, choosing treatments, and planning surgery. Even small errors at the boundaries can lead to overestimation or underestimation of hemorrhage volume, a key factor in predicting mortality and determining whether a patient qualifies for minimally invasive procedures. Missing small or irregular bleeds can delay diagnosis or hide early hematoma growth, while false positives can make the condition seem worse than it is and lead to unnecessary treatments. This shows why it is important to reduce boundary errors (measured by $d_{H95}$) and improve volumetric overlap (measured by $F$-Score and $IoU$), especially for small or changing hemorrhages, where treatment decisions are most affected by segmentation uncertainty.

## 6. Summary and Conclusions

In this work, we investigated the use of CNN variants for IPH segmentation, aligning with current findings on the best techniques in the literature. In essence, we tested statistical models to identify the best CNN variant that accounts for lesion severity and multi-site data heterogeneity.

Our findings demonstrate that Attention U-Net, FPN, and Trans U-Net significantly improve automated IPH segmentation relative to the baseline U-Net, as measured by F-measure and IoU. In contrast, improvements in boundary definition were more selective. Only FPN achieved a statistically significant reduction in boundary error ($d_{H95}$) relative to the baseline U-Net after correction for multiple comparisons, whereas Attention U-Net and Trans U-Net did not show significant boundary improvements. Swin U-Net consistently underperformed relative to the convolutional architectures for both overlap- and boundary-based metrics when compared with Attention U-Net, Trans U-Net, and FPN. Collectively, these results indicate that architectural advances improve volumetric accuracy, but gains in boundary precision are not uniform across models.

Performance improvements were most evident for larger, more confluent hemorrhages, while segmentation of small or irregular IPH remains challenging across all architectures. No significant effects of age, sex, or acquisition site were observed after correction for multiple comparisons, supporting the robustness and generalizability of the evaluated models across heterogeneous clinical imaging conditions in Canada. By enabling more accurate and reliable IPH segmentation across sites, these models have the potential to reduce reliance on labor-intensive manual delineation and streamline acute stroke workflows.

Despite these promising results, several limitations remain. Segmentation of very small or early-stage hemorrhages continues to pose difficulties, suggesting opportunities for future investigation. Additionally, while the present multi-site evaluation supports generalizability within a national context, external validation on geographically and demographically distinct datasets will be essential to confirm broader clinical applicability.

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
