# OpenReview forum: "Multi-site Benchmarking of Deep Learning Models for Intraparenchymal Hemorrhage Segmentation on NCCT"
_MIDL.io/2026/Validation_Papers — MIDL 2026 - Validation Papers Poster_

### Official Review · Reviewer_J641 · 2026-01-03

**Confidence:** 5
**Preliminary Rating:** 3
**Final Rating:** 4

**Summary:**

This study benchmarks four CNN architectures for intraparenchymal hemorrhage  segmentation on NCCT scans using a 17-site dataset of 239 subjects, evaluated via F-measure, IoU, and 95% Hausdorff Distance. Advanced variants outperformed baseline U-Net in volumetric metrics and (for Attention U-Net/FPN) boundary error, showing strong cross-site generalization with minimal site-specific impacts.

**Strengths:**

1. Multi-site validation addresses domain shift from varying scanners/protocols, ensuring robustness in real-world clinical scenarios.
2. IPH-specific optimization excludes mixed ICH subtypes, honing in on the high-mortality subtype’s unique needs and enhancing direct clinical relevance.
3. The approach aligns model performance with clinician priorities, making the findings more actionable than generic segmentation research.

**Weaknesses:**

1. The compared models  are mostly pre-2022 designs, the most recent architectures like nnU-Net v2 and Swin UNETR that have become benchmarks in medical segmentation since 2023.
2. The study fails to incorporate emerging paradigm models (e.g., text-prompt-driven foundation models like SAT, self-supervised learning pipelines).
3. It omits recent models tailored to IPH-specific challenges, reducing the practical guidance value for clinicians seeking deployment-ready, subtype-optimized solutions.

**Detailed Comments:**

1. Section 3.3 focuses on four pre-2022 models and does not include post-2023 state-of-the-art models with clear explanation.
2. Section 2 does not fully cover recent IPH-specific models in the related work.
3. Section 3.4 adopts supervised training and does not incorporate label-efficient self-supervised learning pipelines without specifying the rationale.

**Justification Of Final Rating:**

Thank you to the authors for their response to my previous review comments. Most of my concerns have been addressed, and the paper’s presentation has also been improved. Accordingly, I revise my rating to 4.

**Justification Of The Preliminary Rating:**

The reason for my Borderline recommendation is as I mentioned in the Weaknesses and Detailed Comments: the manuscript’s comparison methods are outdated, making it difficult to support the claims put forward by the authors.

**Questions To Address In The Rebuttal:**

Please give more insight on most recent research as mentioned in Weaknesses and Detailed Comments.

---

### Official Review · Reviewer_MWA9 · 2026-01-09

**Confidence:** 5
**Preliminary Rating:** 3
**Final Rating:** 4

**Summary:**

This paper conducts a multi-site benchmark of U-Net–based models for IPH segmentation on non-contrast CT, focusing on robustness across institutions. While the stratified analyses by age and lesion volume are informative, the study primarily offers empirical comparison rather than methodological innovation, and the exploratory error analyses remain descriptive.

**Strengths:**

- The work focuses on real-world, multi-site data is appropriate for a validation study.
- The use of data from multiple institutions and appropriate patient-level splitting.
- Stratifying performance by age and lesion volume.

**Weaknesses:**

- The benchmarking focuses exclusively on U-Net–family architectures. While these are well-established and widely used, the paper does not clearly explain why the evaluation is restricted to this family.

- The paper relies almost entirely on quantitative metrics, with minimal qualitative visualization of segmentation results.

**Detailed Comments:**

- The authors are encouraged to clarify whether the intent is to validate commonly deployed architectures or to draw broader conclusions about deep learning segmentation robustness would improve interpretability.

- For Figure 2, please consider improving the colorbar and legend.

- A short discussion of how segmentation errors might propagate to downstream tasks would be helpful.

**Justification Of Final Rating:**

The rebuttal clearly and satisfactorily addresses the raised concerns. The authors now provide a well-motivated justification for focusing on U-Net–family architectures, framing the benchmark as a controlled evaluation of commonly deployed segmentation designs rather than a broad survey of all deep learning approaches.

The added qualitative visualizations strengthen interpretability, and the new discussion of how segmentation errors propagate to downstream clinical tasks meaningfully enhances translational relevance. While the study remains intentionally scoped and the qualitative analysis illustrative rather than exhaustive, the revisions improve clarity, positioning, and practical value.

**Justification Of The Preliminary Rating:**

The experimental setup is largely sound and the dataset is clinically relevant, several aspects of the validation scope, qualitative evidence, and result presentation could be strengthened to better meet the expectations of the Validation Studies Track.

**Questions To Address In The Rebuttal:**

Please check the weakness and detailed comments sections.

---

### Official Review · Reviewer_L1DR · 2026-01-09

**Confidence:** 5
**Preliminary Rating:** 3

**Summary:**

This paper focuses on the automated segmentation of intracerebral parenchymal hemorrhage (IPH), a highly lethal subtype of stroke. It designs a multi-center benchmarking scheme and systematically compares the performance of four mainstream CNN architectures on heterogeneous datasets from 17 clinical sites. The research topic is closely aligned with clinical needs, the data diversity is a notable strength, and the experimental design is standardized, with fairly comprehensive analysis of the results, providing a valuable reference for the clinical translation of IPH segmentation models. However, there remains room for improvement in terms of model innovation, analysis of performance in specific scenarios, and supplementation of methodological details.

**Strengths:**

1. The core strength of this paper lies in its clear focus on a key bottleneck in the clinical translation of deep learning models: multi-center generalization capability. Rather than pursuing absolute architectural novelty, the research conducts a systematic benchmark test of existing, well-established CNN variants on a significantly heterogeneous private dataset collected from 17 clinical centers.
2. The annotation process followed a semi-automated approach of ‘algorithmic lesion proposal + expert manual correction,’ with diagnostic confirmation by certified neurologists, ensuring high reliability of the ground-truth data.
3. The clinical data processing was meticulous, with preprocessing steps such as skull stripping and intensity normalization optimized for CT image characteristics. A selective slice sampling strategy was employed to address class imbalance issues, and the design of the composite loss function (Dice Loss + Binary Focal Loss) aligns well with the requirements of the segmentation task.
4. A linear regression model was used to control for confounding factors such as age, sex, hemorrhage volume, and study site. Subsequently, ANOVA and post-hoc tests were applied for inter-model comparisons. The statistical methods are rigorous, and the conclusions are reliable.

**Weaknesses:**

1. The study lacks novelty, as it provides no methodological breakthroughs. It merely benchmarks four existing CNN architectures without proposing new model structures, optimization strategies, or cross-domain adaptation methods, resulting in relatively weak methodological novelty.
2. The study lacks timeliness, as it does not compare the impact of different backbones (such as ResNet, EfficientNet) on performance, nor does it contrast with more recent general-purpose medical segmentation models (e.g., nnUnet, MedSAM, etc.).
3. Although the study included data from multiple centers and employed a selective patch strategy, there was a substantial imbalance in sample sizes across sites within the dataset (e.g., site B and Q had only 2 samples each, while site A had 54). Although the linear model incorporated site as a fixed effect, the reliability of results for sites with extremely small sample sizes should be interpreted with caution. I would expect a more in-depth discussion of this limitation.

**Detailed Comments:**

The study only mentions that segmenting very small or early-stage hemorrhages remains challenging, without quantifying the differences in model performance across different hemorrhage volume ranges (e.g., < 5 mL, 5–30 mL, > 30 mL) or analyzing the specific causes of segmentation errors for small lesions. It also does not discuss the model’s performance in anatomically complex hemorrhage scenarios, such as IPH near sulci or ventricles, which are common in clinical practice and present greater segmentation difficulty. Indeed, this reflects a broader challenge in medical artificial intelligence: balancing specificity and generalizability. I continue to support and encourage researchers to engage in in‑depth exploration of such nuanced research questions.

**Justification Of The Preliminary Rating:**

The preliminary rating of ’Borderline‘ for this manuscript is justified based on a comprehensive assessment of its scientific merit, clinical relevance, methodological rigor, and existing limitations. The work addresses a critical unmet need in clinical practice but falls short of meeting the standards for unconditional acceptance due to notable gaps in innovation, analysis depth, and clinical translation support.

**Questions To Address In The Rebuttal:**

1. The specific value of the "β balancing coefficient" in data preprocessing is not clearly stated, with only a general description as a balancing coefficient, which affects the reproducibility of the experiment.
2. The learning rate reduction strategy during model training is not described in detail, such as the patience parameter and reduction ratio if using ReduceLROnPlateau.
3. The impact of imbalanced sample size distribution (e.g., 54 cases at site A vs. only 2 cases at sites B and Q) is not discussed, and the reliability of results from small-sample sites remains questionable.

---

### Author Rebuttal · Authors · 2026-01-25

**Rebuttal:**

We thank reviewers for their constructive comments. We respond to major themes and list changes. Access to the revised paper in Supporting Material.

Scope & novelty -   We acknowledge that the study does not propose a new architecture by design. Our intent was to conduct a rigorous, controlled benchmark of U-Net–family variants to isolate the impact of architectural choices (attention, multi-scale fusion, transformer-style context) on IPH segmentation across sites, while avoiding confounds from large-scale pretraining or prompt engineering.

Model selection & recent work -  We added Swin-UNET and expanded Section 2 to cite recent IPH work. These additions clarify trends: continued utility of U-Net variants, growth of multi-task/prognosis pipelines, and early exploration of foundation-model–assisted workflows. We also explain why foundation models and self-supervised pipelines were excluded from this controlled, supervised evaluation.

Multi-site imbalance & revised analysis -  We agree that sample-size imbalance is a limitation. We now use two linear models stratified by site size. Model 1 pools small-n sites (n<15) to assess severity-dependent trends. Model 2 includes larger sites (n > 15) to estimate stable site effects. The Discussion emphasizes cautious interpretation for small sites and calls for balanced multi-center data.

Volume, anatomy, and clinical impact -  We quantified performance by IPH volume ranges (new Fig. 4) and redesigned Fig. 2. New Section 3.6 reports anatomy-stratified results: cortical-adjacent bleeds are harder to segment due to partial-volume effects; periventricular hemorrhages show more consistent contrast and performance. We improved qualitative visuals (revised Fig. 1) and added a concise discussion on how boundary and volumetric errors affect downstream tasks (volume estimation, midline shift, longitudinal tracking).

Reproducibility & reporting -  We specified preprocessing and training details (β=1), the optimizer (Adam), and the learning rate schedule, and will release the code on GitHub upon acceptance.  By adding a recent transformer model, refining statistical analyses, stratifying by volume and anatomy, improving qualitative figures, and broadening related work, the revised manuscript directly addresses the reviewers' concerns and clarifies the contribution: a focused, reproducible assessment of how architectural-level choices affect IPH segmentation across clinical sites. We appreciate the reviewers’ guidance.

**Supporting Material:**

/attachment/27cae59cf486a72f92cfc01265a057e865f82919.pdf

---

### Meta-Review · Area_Chair_A48h · 2026-02-05

**Recommendation:** Accept (Poster)
**Confidence:** 4

**Metareview:**

This work provides a rigorous multi-center benchmark of U-Net-based deep learning models for segmenting intracerebral hemorrhage in CT scans, systematically evaluating their robustness across diverse clinical sites to inform reliable clinical translation. Its significance is high due to its direct clinical focus on a lethal stroke subtype and its systematic evaluation of generalization across 17 heterogeneous centers. The clarity and quality are substantially strengthened by the comprehensive rebuttal. The authors convincingly addressed core concerns: they enhanced timeliness by adding Swin U-Net to the benchmark; revised the statistical analysis to stratify by site sample size, thereby responsibly handling data imbalance; and enriched the clinical relevance with new analyses stratified by hemorrhage volume and anatomical location. Additional improvements include better qualitative visualizations and a discussion of error propagation to downstream tasks. The reviewers' updated scores reflect that the rebuttal successfully resolved major concerns, solidifying the work's value as a robust, clinically grounded reference study, thus leaning toward acceptance.

---

### Decision · Program_Chairs · 2026-02-14

Accept (Poster)